# Effects of Augmented Reality-Based Dual-Task Program on Physical Ability by Cognitive Stage with Developmental Disabilities

**DOI:** 10.3390/healthcare10102067

**Published:** 2022-10-18

**Authors:** Hye-Yun Kang, Dong-Yeop Lee, Ji-Heon Hong, Jin-Seop Kim, Seong-Gil Kim, Yong-Gon Seo, Jae-Ho Yu

**Affiliations:** 1Department of Physical Therapy, Sunmoon University, Asan 31460, Korea; 2Division of Sports Medicine, Department of Orthopedic Surgery, Samsung Medical Center, Seoul 06351, Korea

**Keywords:** developmental disability, dual-task, augmented reality, physical ability, cognitive stage

## Abstract

People diagnosed with developmental disabilities are less likely to participate in physical activities even if they are provided opportunities. This study aimed to examine the effects of dual-task exercise-based augmented reality (AR) on muscle strength, muscle endurance, balance ability, and flexibility among people with developmental disabilities. Twenty-seven patients with developmental disabilities were included in the study. The intervention was based on an AR-based rehabilitation program and lasted for 8 weeks. The results showed a statistically significant improvement in muscle strength, muscle endurance, balance ability, and flexibility after the intervention (*p* < 0.05). The AR-based dual-task program increased interest and motivation in the high-cognitive-stage groups, while less interest and motivation were observed in the low-cognitive-stage groups. Our results suggest that an AR-based dual-task program can be an effective method to improve physical ability in patients with high cognitive levels.

## 1. Introduction

Developmental disabilities (DD) refer to delays in two or more developmental areas among children under 5 years of age. Areas for assessing developmental delay include large or small muscle motor skills, speaking, language, cognition, personal–social skills, and activities of daily living [1]. People with DD may be underemployed in terms of athletic capabilities such as balance, coordination, agility, walking, and complexity, and these are significant when the complexity and diversity of the task increase [2]. The manifestation of such developmental delays is often due to language-based self-expression, an impaired physical ability that appears when the body is used, and many other difficulties that result in the lack of social adaptability in sustaining an individual’s activities of daily living or social life [3]. Generally, DD are associated with participation in physical activities for less time [4]. To address this phenomenon, people diagnosed with developmental disorders need a system to alleviate their physical disabilities and cognitive dysfunction [5].

The motor abilities gained through healthy development have a great impact on improving the cognitive abilities of people with developmental disabilities and daily activities and usually serve as a stepping stone for learning more complex movements and task performance skills [6]. Solving complex and different types of problems requires considerable attention, and solving more difficult problems requires an even increased amount of attention. People with DD can upset their balance while solving problems during walking, with increased movement of body parts producing larger and faster movement of the center of mass [7]. Dual-task training for people with DD is associated with a decreased imbalance and increased postural stability, which indicates that the movement path of the center of mass is longer [8].

In general, daily activities require the performance of accurate cognitive tasks in children while being physically capable of maintaining suppleness and coordination. Therefore, dual tasks are an important factor for a healthy life and in improving health conditions in patients with decreased cognition [9]. A previous study [10] reported the positive benefits of dual tasks in improving independent daily living skills and problem-solving capacities. Dual task refers to the performance of two tasks simultaneously. It is generally used to describe a combination of cognitive and motor tasks (i.e., walking while counting numbers) and implemented to improve one or both functions. Considering that individuals with DD have different disabilities and symptoms, exercise plans are required to target their specific health conditions. Therefore, it remains a challenge to implement exercise plans for individuals at a low cognitive stage. Their cognitive stages are related to planning total motor and microscopic motor abilities, and it has been reported that the total number of motor abilities can lead to improvements in cognitive and social skills for those with DD [11].

Previous studies [12] have examined various intervention methods to improve the physical abilities of people with DD, and a few studies have been conducted using dual tasks or intervention programs based on enhanced reality. In the past decades, the use of augmented reality (AR) in rehabilitation has increased and many interactive training contents have been made accessible. In a systematic article by Garzón [13], several benefits of using AR-based training programs were reported for people with DD. AR as an intervention can be used to perform a variety of physical activities, improve physical abilities, and increase the rate of functional skill learning [14]. Cakir and Korkmaz [15] emphasized that AR-based programs increase independent living skills and the special skills needed for activities of daily living. However, studies on the use of enhanced realistic techniques are increasing owing to physical therapeutic interventions, but the scale of such studies is small, heterogeneous, and lacks control. The results of these studies lack proper conclusions; therefore, it is difficult to evaluate their advantages compared to existing methods [16].

The purpose of this study was to examine the effects of AR-based dual-task exercise on physical abilities such as muscle strength, muscle endurance, balance ability, and flexibility among people with DD.

## 2. Materials and Methods

### 2.1. Study Design

This study comprised pre- and post-test designs. Participants were divided into three groups depending on the cognitive stage—infant stage group (ISG), childhood stage group (CSG), and adult stage group (ASG). The intervention proceeded from ISG on the lower difficulty level to CSG and ASG with increasing difficulty levels.

### 2.2. Study Participants

Twenty-seven patients with DD selected from the Disabled Persons Week Conservation Center in Chung Cheong Nam-do participated in this study. This study was approved by the Institutional Review Board of Sunmoon University (SM-202112-073-2). The exclusion criteria were: (1) patients diagnosed with intellectual disabilities other than general developmental disorders, cerebral palsy, chromosomal disorders, attention deficit and hyper-behavior disorders, and developmental disorders due to autism; (2) patients with general DD with delay disorders; and (3) patients with DD who could not understand and follow the instructions given by researchers.

### 2.3. Experimental Procedure

To classify the groups according to the subject’s cognitive stage, cognitive and physical evaluations were performed in the pre-test, and only physical evaluation was performed in the post-test. This research was conducted in the following steps (Figure 1).

#### Augmented Reality-Based Dual Task

The dual-task program was conducted using AR-based rehabilitation program equipment UINCARE-82B, Seoul, Korea, (Figure 2A) with the UINHealth PRO ver. 2 software. The UINHealth PRO ver. 2 includes various rehabilitation exercises which target principally physical and cognitive functions. An illustration of a participant performing the exercise is shown in Figure 2B. The exercises used in the present study, with their respective performance times, are shown in Table 1 below. The subjects conducted the exercise program once a week for 8 weeks. One set took 15 min to participate in one exercise program, and a total of two sets were carried out. The duration of the exercise was 40 min, including rest time. The difficulty levels in each group were adjusted by setting “high, medium, and low” levels installed on the device for the AR rehabilitation programs.

### 2.4. Measurements and Instrumentation

#### 2.4.1. Cognitive Stage

Cognitive stage was assessed using the Computer Cognitive Senior Assessment System-Screen (CoSAS-S), according to a previous study [17], which was effective in cognitive assessment of persons with DD. A set of 29 questions was used to assess the cognitive stage based on memory, concentration, visual ability, etc. Enter the current year, match the location of the target person, and look for the wrong part of the image displayed on the left side and the image displayed on the right side of the screen, displayed at the end rather than waiting for the moving point. This consists of searching for the location and matching the number of stacked blocks. Based on the results, 0 to 31.12 points were classified as ISG, 31.13 to 62.23 points as CSG, and 62.24 to 100 points as ASG.

#### 2.4.2. Physical Ability

Physical ability, including muscle strength, muscle endurance, balance, and flexibility, were all assessed using AR-based rehabilitation program equipment (UINCARE, Seoul, Korea, 82-B, Figure 2).

(1)Muscle Strength

Participants begin the exercises in a sitting position on a chair with a backrest after performing a sensor initialization at two meters from the device. The arms were raised above the chest in the shape of X. After pressing the start evaluation button, the subject evaluated how many times he stood up and sat back on the chair with the knee fully extended along the signal displayed on the screen.

(2)Muscle Endurance

The muscle endurance was assessed through the 2 min walking test and conducted via the device. After the initialization of the sensor, participants were instructed to walk around for 2 min, and the walking distance was recorded and saved for analysis. 

(3)Balance

The Time up and go test was used to assess the balance ability. The chair was placed three meters away from the device and the subject started in a sitting position. After pressing the evaluation start button, the subject stood up from the chair, walked three meters, and touched the device monitor. Subsequently, the time required to return to the chair and sit down was measured.

(4)Flexibility

The flexibility of the upper and lower extremities was assessed in a sitting position. First, for the upper extremity, the shoulder joints were in 90° abduction, the elbow joints were in 90° flexion, and shoulder external rotations and internal rotations were performed to assess the flexibility of the upper limbs. The flexibility of the lower limbs was evaluated by flexion and extension of the knee.

### 2.5. Data Analysis

Statistical analyses were performed using SPSS ver. 26.0 (IBM Corporation, Armonk, NY, USA) for Windows, and all the statistics were expressed as means and standard deviations. Descriptive statistics were used to analyze the general characteristics of the participants. Parametric statistics were used because the data in this study met the normality test. A paired *t-*test was performed to confirm the difference between pre- and post-intervention values within the groups. One-way ANOVA was conducted to analyze the differences between the groups. The Bonferroni test was used as the post hoc test when there were differences between the groups, and all significance levels were set at *p* < 0.05.

## 3. Results

### 3.1. General Characteristics of Subjects

Twenty children diagnosed with a DD were recruited, while three were excluded before the pre-test, one due to autistic symptoms and the two others due to transportation issues. A total of 27 people with DD participated in this study. The general characteristics classified by the cognitive stage are presented in Table 2. When CoSAS-S evaluated the cognitive stage, 0 to 31.12 points were classified as ISG, 32.13 to 62.23 points as CSG, and 62.24 to 100 points as ASG.

### 3.2. Comparison of Differences between AR-Based Dual-Task Programs

Table 3 presents the results of the intervention for each stage. In the ISG, muscle strength, muscle endurance, balance, and flexibility showed statistically significant changes (*p* = 0.01, *p* = 0.005, *p* = 0.008, and *p* = 0.010, respectively). For the CSG, muscle strength, balance, and flexibility showed significant increases (*p* = 0.000, *p* = 0.001, and *p* = 0.000, respectively). No significant improvement was observed in muscle endurance (*p* = 0.065). In the ASG, significant improvement was observed in muscle strength, muscle endurance, balance, and flexibility (*p* = 0.000, *p* = 0.000, *p* = 0.009, and *p* = 0.000, respectively).

### 3.3. Comparison between Groups of the Difference Value after Intervention and before Intervention

There were no statistically significant differences in muscle strength, muscular earth strength, or balance between the groups (*p* > 0.05), but there was a statistically significant difference in flexibility (*p* < 0.01). In the Bonferroni post hoc test for flexibility, the results showed a significant difference between the ISG and ASG (*p* < 0.05) and between the CSG and ASG (*p* < 0.05) groups (Table 4), (Figure 3).

## 4. Discussion

This study aimed to evaluate the effects of an AR-based dual-task program on the physical abilities of people with DD. Our results demonstrated that the intervention using an AR-based dual task may be beneficial in improving the physical abilities of people diagnosed with DD.

People with DD may need more cognitive resources because of the lower level of automation in cognitive processing, which may lead to cognitive overload and insufficient attention [18]. The success of a dual task depends on the level of automation of the cognitive processing process. It is known that when attention is focused on the outside, a more naturally self-organized motor process, including an automatic control mechanism, occurs. This leads to a conscious type of control, as described by the limited behavioral hypothesis, and shows better motor performance than work focused on limiting motor performance [19]. An explanation supporting the improved motor performance of participants after performing dual tasks is their increased ability to control their body movements. Given the limited time available to coordinate primary body tasks while simultaneously performing secondary cognitive tasks, participants were required to make rapid body adjustments guided by visual feedback to achieve goal-directed movements. This explanation is related to the use of the forward mechanism. Therefore, the performance of a cognitive task using a visual stimulus can be temporally pressured for rapid physical adjustment and to promote the automation of physical adjustment [20]. Thus, dual-task training for the improvement of physical abilities can promote physical automation in individuals with DD. This is explained by the improvement in motor performance, which is considered an effective measure to reduce cognitive overload.

In this study, the AR-based dual-task program improved physical activity in patients with DD. This can be seen as reducing the cognitive overload phenomenon by practicing a cognitive task that may occur while performing a secondary cognitive task through intervention [21]. Patients with DD are accustomed to touching real objects and receiving feedback. The AR-based training program used in the present study was able to provide feedback on user movements using a Kinect sensor, which tracked the movement of the body and provided feedback [22].

Muscle endurance had a significant effect on ISG and ASG but not on CSG. This result suggests that the transition from ISG to CSG is an important phase when the development of physical strength and obesity occurs. Men tend to gain more physical strength, whereas women tend to stagnate in the adolescent stage of typical development [23]. These developmental patterns have led to significant gender differences among adolescents with DD. With low levels of physical strength, adipose tissue increases at a higher rate among women during adolescence [24]. However, this study was conducted without considering the proportion of the genders in each group; therefore, it can be assumed that the proportion of women in the childhood stage group was high.

In this study, flexibility was observed as a significant difference between the groups. Additionally, during the post-test, the subject would have become familiar with the device and accepted the feedback provided during the initial evaluation and intervention, resulting in correct posture and position. These results suggest the importance of maintaining correct posture during the pre- and post-tests.

The results of this study showed that among the variables, only flexibility matched the hypothesis, and the higher the cognitive stage, the higher the improvement in physical ability. According to previous studies, cognitive stages similar to those of their normally developed peers show higher rates of improvement in language and motor abilities. This phenomenon is related not only to physical age but also to cognitive age [22]. Furthermore, groups at low cognitive stages exhibit low concentration and characteristics of boredom in the evaluation, which makes evaluation difficult. For this reason, low evaluation scores were observed during the pre-test in those at the low cognitive stage while high evaluation scores were observed in those at the high cognitive stage. Therefore, deriving results that did not match the hypothesis of this study in terms of muscle strength, muscular endurance, and balance was inevitable.

This study has some limitations. First, it is difficult to conclude that this is an advanced study without a control group and that an AR-based dual-task program is the best way to improve the physical fitness of people with DD. Second, there are difficulties in conducting research on people with DD who have been diagnosed with general developmental disorders and generalizing them as effective methods for all types of people with this issue. Third, it cannot be generalized whether all physical abilities that can be achieved are not set as dependent eccentrics. The research conducted by selecting the four representatives improves all physical abilities. Finally, it was difficult to re-exercise the disinterested subjects. Therefore, in future research, it is necessary to supplement these restrictions and target various types of developmental disorders and include control groups to proceed with the research.

## 5. Conclusions

The study demonstrated that a realistic dual-task exercise is associated with improvement in overall physical abilities, including muscle strength, muscle endurance, balance, and flexibility, in people with DD. Therefore, an AR-based dual-task program can be considered a selective intervention method for improving the physical abilities of people with DD.

## Figures and Tables

**Figure 1 healthcare-10-02067-f001:**
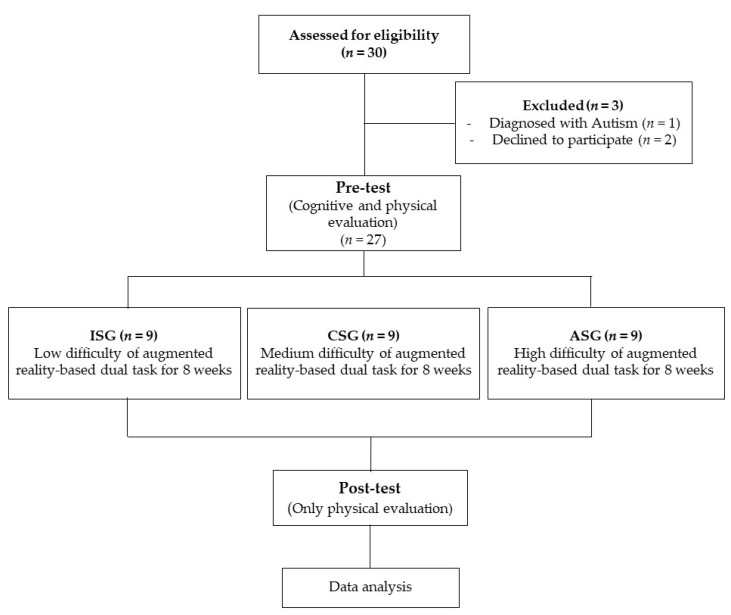
Experimental procedure.

**Figure 2 healthcare-10-02067-f002:**
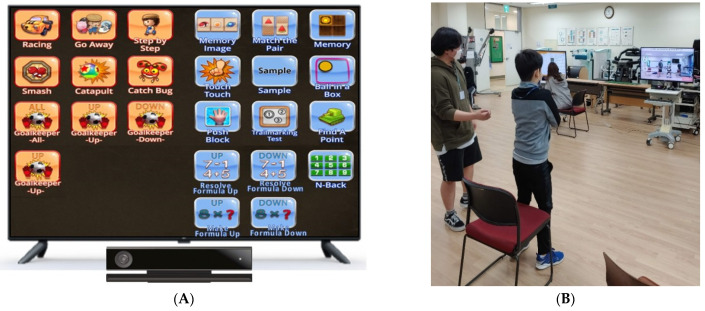
(**A**) Augmented reality-based rehabilitation device. (**B**) Participant during the exercise session.

**Figure 3 healthcare-10-02067-f003:**
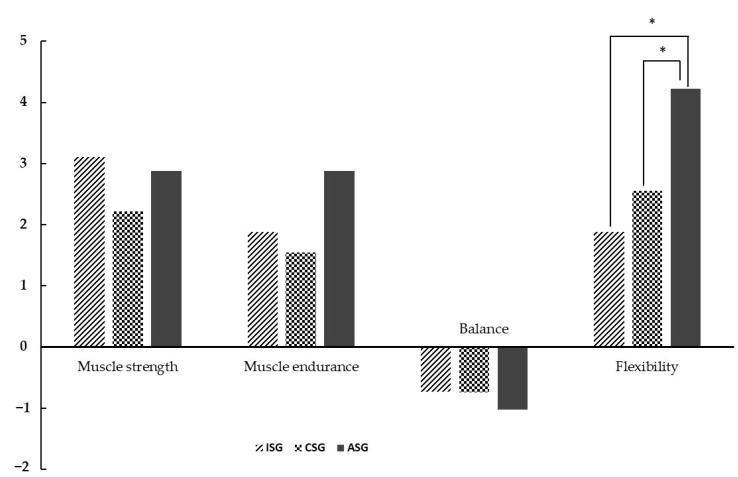
Comparison of the results between groups. ISG = infant stage group, CSG = childhood stage group, ASG = adult stage group. Muscle strength, muscle endurance (number of repetitions), balance (seconds), and flexibility (degrees), * *p* < 0.05.

**Table 1 healthcare-10-02067-t001:** Details and times of dual-task program.

Exercise Name	Exercise Goal	Time (min)
Catch Bug	The goal is to catch moving insects using shoulder flexion, adduction, and abduction to get high scores at a set time.	2
	Low level: 1 insect appears every 3 s, and a group of insects appears every 30 s	
	Medium level: 1 insect appears every 2 s, and a group of insects appears every 30 s	
	High level: 1 insect appears every 1 s, and a group of insects appears every 30 s	
Go Away	The goal is to defeat the monster who attacks the target using shoulder adduction, abduction, and flexion without being attacked at a set time.	2
	Low level: Two monsters appear at a slow pace once every 3 s.	
	Medium level: Two monsters appear at a medium pace once every 3 s.	
	High level: Two monsters appear at a fast pace once every 3 s.	
Memory Image	A picture to remember appears on the screen for 3 s; memorize and find the one that appears early on the over 16-picture display on the screen. Required to maintain shoulder position for 3 s to achieve the task.	2
	Low level: Remember one picture.	
	Medium level: Remember two pictures.	
	High level: Remember three pictures.	
Trail-Making	Connect the different numbers between them according to the ascending or descending order.	2
	Low level: from 1 to 10	
	Medium level: from 1 to 15	
	High level: from 1 to 20	
Goalkeeper	Block the soccer ball with your leg or arm depending on where the ball passes.	2
	Low level: estimated time of ball position 5 s	
	Medium level: estimated time of ball position 4 s	
	High level: estimated time of ball position 3 s	
Racing	Use your head and trunk to avoid obstacles when driving a car during a race and collect more gifts to have a higher score.	3
	Low level: a small amount of obstruction	
	Medium level: moderate amount of obstruction	
	High level: the amount of obstruction	
Calculate Formula	To solve the equation, keep your hand on the appropriate computational symbol for 3 s.	2
	Low level: Calculate a formula with a single digit.	
	Medium level: Calculate a formula with a double-digit number.	
	High level: Calculate a formula with a three-digit number.	

**Table 2 healthcare-10-02067-t002:** General characteristics in participants.

Variables	Infant Stage (*n* = 9)	Childhood Stage (*n* = 9)	Adult Stage (*n* = 9)
Age (years)	11.77 ± 3.39	14.33 ± 2.86	16.55 ± 2.83
Height (cm)	135.9 ± 20.90	146.17 ± 25.86	152.6 ± 15.53
Weight (kg)	47.7 ± 11.64	52.1 ± 18.69	55.87 ± 12.15
CoSAS-S	16.82 ± 7.09	45.45 ± 9.04	71.06 ± 8.57

Mean ± standard deviation. CoSAS-S; Computer Cognitive Senior Assessment System-Screen.

**Table 3 healthcare-10-02067-t003:** Differences before and after the augmented reality-based intervention in each group.

Stage	Variables	Pre-Intervention	Post-Intervention	*p*
Infant stage (*n* = 9)	Muscle Strength (number of times)	11.00 ± 1.50	14.11 ± 1.69 **	0.001
	Muscle Endurance (number of times)	21.78 ± 3.86	23.67 ± 3.42 **	0.005
	Balance (second)	17.08 ± 4.85	16.35 ± 4.43 **	0.008
	Flexibility (degree)	14.00 ± 3.00	15.89 ± 2.75 *	0.010
Childhood stage (*n* = 9)	Muscle Strength (number of times)	31.22 ± 5.95	34.56 ± 5.61 ***	0.000
	Muscle Endurance (number of times)	31.44 ± 2.45	33.00 ± 3.12	0.065
	Balance (second)	13.64 ± 0.83	12.90 ± 0.60 **	0.001
	Flexibility (degree)	23.44 ± 2.18	26.00 ± 2.12 ***	0.000
Adult stage (*n* = 9)	Muscle Strength (number of times)	31.44 ± 2.06	34.33 ± 1.58 ***	0.000
	Muscle Endurance (number of times)	41.00 ± 4.03	43.89 ± 4.40 ***	0.000
	Balance (second)	13.74 ± 1.47	12.72 ± 0.84 **	0.009
	Flexibility (degree)	28.67 ± 4.77	32.89 ± 4.34 ***	0.000

Values are mean ± standard deviation. *p*-value is described before and after intervention in each group (paired *t*-test) * *p* < 0.05, ** *p* < 0.01, *** *p* < 0.001.

**Table 4 healthcare-10-02067-t004:** Differences in physical ability between the groups.

Variables	ISG (*n* = 9)	CSG (*n* = 9)	ASG (*n* = 9)	F	*p*
Muscle Strength (number of times)	3.11 ± 1.76	3.33 ± 1.11	2.88 ± 1.16	0.233	0.794
Muscle Endurance (number of times)	1.88 ± 1.45	1.55 ± 2.18	2.88 ± 1.05	1.625	0.218
Balance (seconds)	−0.73 ± 0.63	−0.74 ± 0.43	−1.02 ± 0.88	0.524	0.599
Flexibility (degrees)	1.88 ± 1.69	2.55 ± 0.52	4.22 ± 1.20	8.509	0.002 **

Values are mean ± standard deviation. *p*-value is described before and after intervention between the groups (one-way ANOVA), ** *p* < 0.01.

## Data Availability

The data used to support the findings of this study are available from the corresponding author upon reasonable request.

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
