# Peer review of "Effects of Augmented Reality-Based Dual-Task Program on Physical Ability by Cognitive Stage with Developmental Disabilities"

_healthcare, 2022, doi:10.3390/healthcare10102067_

Round 1

Reviewer 1 Report

Dears Authors

The study is relevant; however, several points need to be improved and deepened.

The state of the art is deficient, the justification needs to be better worked out, as well as the presentation of the results.

The rationale of the study presented in the "Introduction" section needs to be further elaborated.

Figures 2 and 3 must be replaced, to their low resolution (Figure 2), and difficulty to read the values (Figure 3)

Reviewer 2 Report

This paper describes a study on the effects of an AR-based dual task program on the physical ability of the cognitive stage with developmental disabilities (DD). It is well written with proper structure. The language and content need revisions to improve its readability. Try to make the sentences short by reducing ", and" to make the article concise. Some details in in Section 2 are needed to make them clear.

Specifically,

Line 18.  ….program and lasted for…. Delete the “and”.

Lines 38-40. This paragraph contains only two sentences. It is too short. Consider combining these sentences into the previous paragraph in a smooth manner.

Line 60. “Many previous studies…” How many?  Delete “Many”. It is redundant.

Line 72. “Therefore, the purpose of this study…”. Delete “Therefore,” it is redundant.

Line 98. Is the UINCARE, Korea, 82-B a hardware or a software?? Is it the one in Figure 2?? You need to show a figure here when you mention it the first time. If it is hardware, then you need to indicate the software. 

Table 1. There are seven exercises in this table. Figure 2 shows a displayer on a stand. Were these exercises shown in this displayer? or in some other device?

It is difficult for the readers to understand what were they look like or what were the participants doing with those exercises. Try to show some photos of these exercises to make them more clear.

Line 111. Change “, etc…” to “, etc.”

Section 2.4.2. You mentioned “the AR-based rehabilitation equipment (UINCARE, Korea, 82-B)” five times in this section. It is very wordy. Please mentioned only one time (in the beginning of the section) to make it concise.  

Figure 3. Please put a vertical axis with a title. You need to indicate the units for each variable.

Section 3.1 There were 27 participants. What were their gender?  You didn't consider gender. Why? Did you have evidence that gender is not a significant factor affecting the variables? 

Line 231. Change “….; therefore, it can be….” to “. Therefore, it can be…”

Line 244. “During the pre-test, for this reason, low evaluation scores…..at the high cognitive stage.” This sentence is problematic. Please rewrite it.

Line 255. Change “…eccentrics, and the research conducted…” to “..eccentrics. The research conducted…”

Reviewer 3 Report

The article sounds very stimulating and makes interesting observations in the area of ​​using AR for people with DD.

Within the article, please correct the following formal parts:

In references, correct the order of name and surname, in ref. 12 you state the surname in capital letters, in some references you state abbreviations of names and sometimes full names. Please edit based on journal requirements.

Page 6 is empty, please remove it.

Figure 1 - increase the width of the image, it is too narrow and edit the word ELIGIBILITY in the first box.

Figure 2 - if possible, replace the image with a higher quality (resolution) image.

Figure 3 - if possible, add a scale to the graphs and enlarge the numbers for readability.

Table 1 - put the divided words, especially in the first column, under each other.

It would be appropriate to add an Image of participant while using AR equipment to the article.

As part of the future evaluation, it would be appropriate to focus on the assessment of the increase in the quality of life of the participants after passing the given tests. There are various types of subjective as well as objectified questionnaires.

The whole team did a good job!

Round 2

Reviewer 1 Report

Dear Authors

I believe that all comments on the manuscript have been heeded!

Regards

Author Response

We sincerely thank you for your efforts for our manuscript.

Reviewer 2 Report

Typical AR is to superimpose the virtual image on the real vision via a headpiece or a AR glasses. In Figure 2, a subject was doing an exercise while watching the TV monitor. How come  the device is an AR-based device? Why was the exercise AR-related?

Line 66, "..augmented reality (A) in ..." should be "..augmented reality (AR) in ...

Author Response

We sincerely thank you for your efforts for our manuscript. According to your comments, we have answered your question and have revised the manuscript. Sincerely.
